

# TSFF: a two-stage fusion framework for 3D object detection

Guoqing Jiang, Saiya Li, Ziyu Huang, Guorong Cai and Jinhe Su

The School of Computer Engineering, Jimei University, Xiamen, Fujian, China

## ABSTRACT

Point clouds are highly regarded in the field of 3D object detection for their superior geometric properties and versatility. However, object occlusion and defects in scanning equipment frequently result in sparse and missing data within point clouds, adversely affecting the final prediction. Recognizing the synergistic potential between the rich semantic information present in images and the geometric data in point clouds for scene representation, we introduce a two-stage fusion framework (TSFF) for 3D object detection. To address the issue of corrupted geometric information in point clouds caused by object occlusion, we augment point features with image features, thereby enhancing the reference factor of the point cloud during the voting bias phase. Furthermore, we implement a constrained fusion module to selectively sample voting points using a 2D bounding box, integrating valuable image features while reducing the impact of background points in sparse scenes. Our methodology was evaluated on the SUNRGB-D dataset, where it achieved a 3.6 mean average percent (mAP) improvement in the mAP@0.25 evaluation criterion over the baseline. In comparison to other great 3D object detection methods, our method had excellent performance in the detection of some objects.

## INTRODUCTION

The advancement of computer vision has significantly enhanced various aspects of human life, with 3D object detection emerging as a crucial subfield with widespread applications in autonomous driving, assistive robotics, and numerous other tasks. The goal of 3D object detection is to localize and identify objects within a scene, which necessitates efficient scene understanding. Conventional research approach involves using RGB images as input (*Guizilini et al., 2020*; *Chen et al., 2021*; *Wang et al., 2021b*; *Huang et al., 2022a*; *Liu et al., 2022*; *Zhang et al., 2023a*; *Wang et al., 2022c*; *Li et al., 2023*). These methods use depth estimation to compute depth information from RGB images, thereby simulating the spatial coordinates of pixels in the 3D space to assist in 3D detection tasks. Although RGB images provide rich texture and semantic information, 3D object detection emphasizes the need for spatial depth, and the lack of depth information significantly impairs the effectiveness of RGB images in 3D detection tasks. Compared to RGB images, point cloud data preserves the geometric structure of objects in 3D space, with each point's 3D coordinates representing the corresponding depth information. Recent 3D object detection tasks primarily use point clouds as input (*Qi et al., 2019*; *Wang et al., 2021a*; *Liu et al., 2021*; *Zhang et al., 2023b*; *Huang et al., 2022b*; *Hu et al., 2023*), utilizing the geometric features of

Corresponding author
Jinhe Su, sujh@jmu.edu.cn

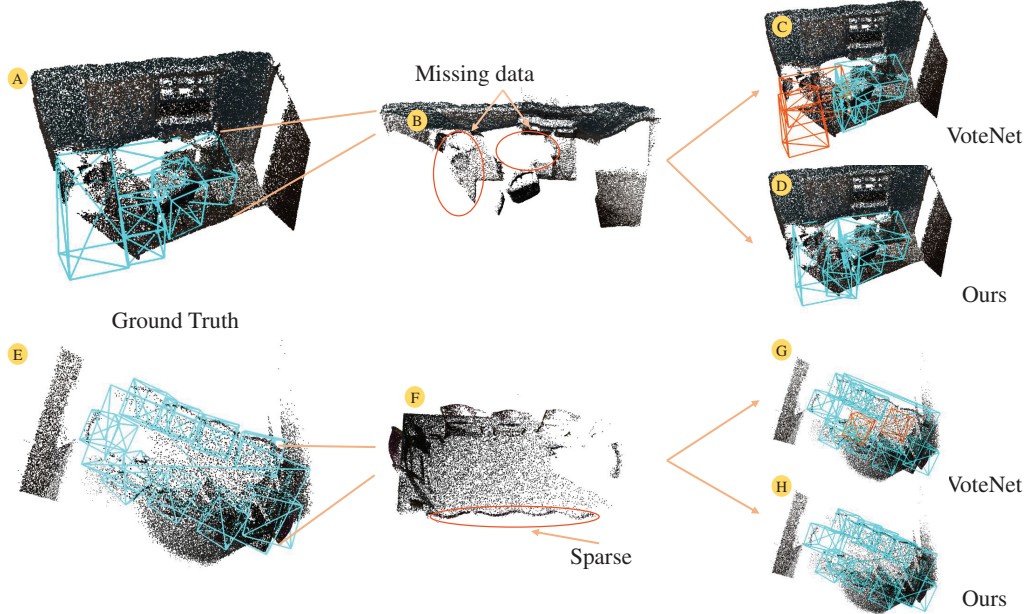

**Figure 1 The 3D scene under the point cloud suffers from sparse and missing data under the influence of object occlusion and scanning equipment.** Facing point clouds with corrupted geometry, the prediction ability of VoteNet for object categories is insufficient (C, G). In contrast, our proposed method demonstrates superior performance (D, H). (A, E) Ground truth for both scenes. For a more intuitive representation of the point cloud scene, the background points of the floor (B, F) are optimized and viewed from the top. Figure source credit: SUN RGB-D dataset.

3D data to achieve outstanding detection results. However, due to the limitations of data acquisition methods and sensor errors, the sparsity of point clouds and data loss pose significant challenges to 3D object detection tasks using point clouds.

Figure 1 illustrates two indoor point cloud scenes. The missing points in the point cloud lead to significant empty areas on the table surface (see Fig. 1B), severely compromising the geometric integrity of the table. Additionally, mutual occlusion between objects results in sparse point clouds of target objects (see Fig. 1F). VoteNet (*Qi et al., 2019*) attempts to address this by using point clouds as input, voting to cluster points towards object centers and selecting object centers through a region aggregation module. However, the aggregation mechanism often fails to handle neighboring object centers properly, leading to erroneous predictions (see Figs. 1C, 1G). Some methods (*Cheng et al., 2021*; *Yu et al., 2023*; *Wang et al., 2022a*) have tried to address feature sparsity by leveraging geometric relationships in surrounding point clouds on top of voting. Nonetheless, the inherent lack of semantic information in point cloud data limits their ability to capture the relationships between objects within a scene. Early studies (*Ren & Sudderth, 2016*; *Lahoud & Ghanem, 2017*; *Song & Xiao, 2016*) attempted to integrate semantic features from RGB images with point features, but the coarse feature matching resulted in suboptimal detection performance. Cross-modal feature fusion has become a popular research topic in 3D object detection as natural language processing has achieved success in image and text features. PointFusion (*Xu, Anguelov & Jain, 2018*) employs two branches to separately process RGB

images and point cloud data, using a fusion network to directly predict object positions and categories. EPNet (*Huang et al., 2020*) attempts to achieve point-wise fusion of image features and point features using point-guided methods. However, the sparsity of point clouds inherently limits the utilization of image features, and the distinct characteristics of each data modality make effective fusion challenging. Therefore, this article investigates the effective use of image semantic features to enhance point cloud features, aiming to achieve more effective indoor 3D object detection.

In this article, we explore the supplementary role of images in 3D object detection and propose a two-stage fusion framework tailored for this task. Addressing the challenge posed by sparse point clouds on object surfaces, which results in a scarcity of available point features, we introduce a voting fusion module. This module projects seed points onto the 2D image of the corresponding scene using a mapping matrix to obtain the respective pixel coordinate points. Subsequently, the image features at these pixels are extracted and combined with the point features, serving as inputs for the voting stage. Our aim is to leverage the rich semantic features to enhance the voting set by biasing the process towards gathering more points related to the target object. To mitigate the impact of background noise points on the final detection, we designed a constraint fusion module. This module restricts the selection of 3D voting points using the 2D detection frame obtained from the 2D detection head, aiming to retain as many foreground points within the voting set as possible while discarding the background points. Additionally, the bias introduced by the voting operation causes some mapping coordinates to exceed the image boundaries. To address this, we normalize the pixel coordinates of these mapping points, ensuring they fall within the image limits. To further enhance the expressiveness of the point features, we perform an additional round of feature fusion. Subsequently, the secondary fused point features are used as inputs for the sampling aggregation operation to predict the 3D object. Finally, 3D non-maximum suppression (NMS) is employed to select the prediction results with high confidence.

 Our key contributions are as follows:

- In scenarios characterized by sparse available point features, image features are fused with seed point features to ensure that the voting points are more significantly influenced during the bias adjustment phase.
- A constraint fusion module is introduced to refine the selection of voting points and reduce the interference from background points.

## RELATED WORK

3D object detection has advanced due to technological progress in sensor devices. Over the past decade, researchers have employed stereo cameras to capture images enriched with depth information. These images are utilized to estimate the geometric coordinates of pixels in 3D space, facilitating 3D object detection. With the emergence of point cloud data types, detection methodologies in various scenarios have evolved into point-based and voxel-based approaches.

**Point cloud based 3D object detection:** Point clouds possess rich spatial geometric information, which indirectly results in their large data volume. To efficiently process point cloud data from different scenes, they are typically categorized into point-based methods (*Xie et al., 2020*; *You et al., 2022*; *Zhao & Qi, 2022*; *Liang, An & Ma, 2022*; *Duan et al., 2022*; *Wang et al., 2022b*; *Shen et al., 2023*; *Wei et al., 2023*) and voxel-based methods (*Yin, Zhou & Krahenbuhl, 2021*; *Zheng et al., 2022*; *Fan et al., 2022b*, *2022a*; *Wu et al., 2023*; *Fan et al., 2023*) based on their representation forms. Point-based methods use raw point clouds as input. MLCVNet (*Xie et al., 2020*) processes point features enhanced by a self-attention mechanism, performs voting, and integrates multi-scale features to capture scene context. However, this approach overlooks the interference of background noise on the voting points. RGBNet (*Wang et al., 2022b*) addresses the issue of background point interference leading to voting bias and ineffective utilization of surface point clouds by focusing on how the surface geometry of foreground objects aids in the voting grouping. It aggregates point-wise features of object surfaces using uniform rays emitted from clustering centers and employs a foreground-biased sampling strategy to obtain more surface point clouds. Nevertheless, in practice, foreground point clouds are often sparse, resulting in inaccurately angled prediction boxes. CanVote (*You et al., 2022*) decomposes the direct offset during the voting process, constrains the prediction of the target box direction with a local normalized coordinate system, and eliminates erroneous predictions through reverse projection. Voxel-based methods are predominantly used for outdoor open scenes. Unlike the confined environments indoors, the data volume of 3D point clouds in outdoor scenes increases exponentially. Therefore, point clouds are often divided into voxel grids, from which voxel features are extracted to reduce computational overhead. SST (*Fan et al., 2022a*) proposes the single-stride sparse transformer, which addresses information loss due to down-sampling by grouping voxelized point clouds into regions and performing regional shifts, benefiting small object detection. Although regional shifts mitigate down-sampling issues, the limitation in receptive field size leads to significant computational costs. FSD (*Fan et al., 2022b*) introduces a sparse detector that eliminates data redundancy by utilizing temporal information, drastically reducing computational overhead and enabling multi-frame perception. FSDv2 (*Fan et al., 2023*) further extends FSD by integrating 2D instance segmentation into the point cloud layer, achieving a leap in inference speed through the proposed sparse architecture.

**Cross-modal fusion based 3D object detection:** Purely point cloud-based 3D object detection is affected by inherent data deficiencies. Some efforts (*Qi et al., 2018*; *Liang et al., 2019*; *Zhang, Chen & Huang, 2022*; *Zheng et al., 2022*; *Zhao et al., 2023*; *Chen et al., 2023*) attempt to integrate image features into point cloud features. F-PointNet (*Qi et al., 2018*) obtains 2D object proposals from images, elevates them to a 3D perspective using depth information, aligns point features and image features within the region, and predicts 3D bounding boxes. The post-fusion method that directly defines 3D candidate regions based on 2D bounding boxes heavily relies on the detection accuracy of the 2D branch, which is disadvantageous for scenes with significant occlusion. LIF-Seg (*Zhao et al., 2023*) is no longer dependent on a single modality. UNet (*Ronneberger, Fischer & Brox, 2015*) was used to learn the coarse features of point clouds after obtaining concatenated features of point

clouds and images through projection matrices. Leveraging the learned coarse features, it predicts the offset between point cloud and image semantic features for feature alignment and fusion. PiMAE (*Chen et al., 2023*) attempts to mask images and point clouds and learns the mutual relationship between the two modal features through attention mechanisms, but the design of masks also limits the network's utilization of local features. The above methods overlook significant background semantic interference when fusing with image features. In contrast, our TSFF can focus the attention on regions of interest in the network, thereby extracting and utilizing more useful 2D semantic information.

## METHODS

As illustrated in Fig. 2, the two-stage fusion framework (TSFF) model we propose is founded on the Deep Hough Voting framework, which enables reliable detection of object positions and categories with the help of voting operations. In this section, we first introduce the feature extraction methods for both point cloud and image data. Subsequently, considering the unsatisfactory prediction performance of VoteNet on sparse point cloud surfaces, we introduce the voting mapping module and try to enhance the point features with image features. The improvement of the voting bias phase is achieved by assigning semantic features to the seed points at the corresponding 2D image locations. Next, we use the constraint fusion module to eliminate the interference of background noise points and sample the voting points with a 2D bounding box as a restriction to obtain voting points with higher applied weights. At the end of this section, we illustrate the loss design.

### Feature extraction

**Point branch:** Indoor point clouds, unlike their large-scale outdoor counterparts, typically represent scenes with a lower data burden. This allows point-based methods to offer greater efficiency compared to voxel-based approaches when choosing a suitable representation for the point cloud data. To address the inherent redundancy within the raw point cloud, we leverage PointNet++ (*Qi et al., 2017*) as our core network for pre-processing. The input point cloud consists of N (typically N = 20k) points, each with four features. The network first employs four set abstraction (SA) layers to reduce the number of points and extract informative features. Subsequently, two feature propagation (FP) layers perform upsampling and propagate features, resulting in K seed points with enhanced features. Here, K is set to 1,024 and each seed point has F = 256 dimensional features. These refined seed points serve as the foundation for subsequent processing stages.

**Image branch:** RGB images are significantly different from point clouds in terms of data types, although the limitation of dimensionality makes it impossible to obtain geometrical information about objects, the semantic nature between pixel values gives 2D graphs a rich expressive capability. In image branch, we adopt the common Faster-RCNN as the edge detection framework. ResNet-50 is used as the backbone of the detector and image feature extraction is performed on the input image with the help of feature pyramid network (FPN) to obtain the image feature map corresponding to the point cloud data.

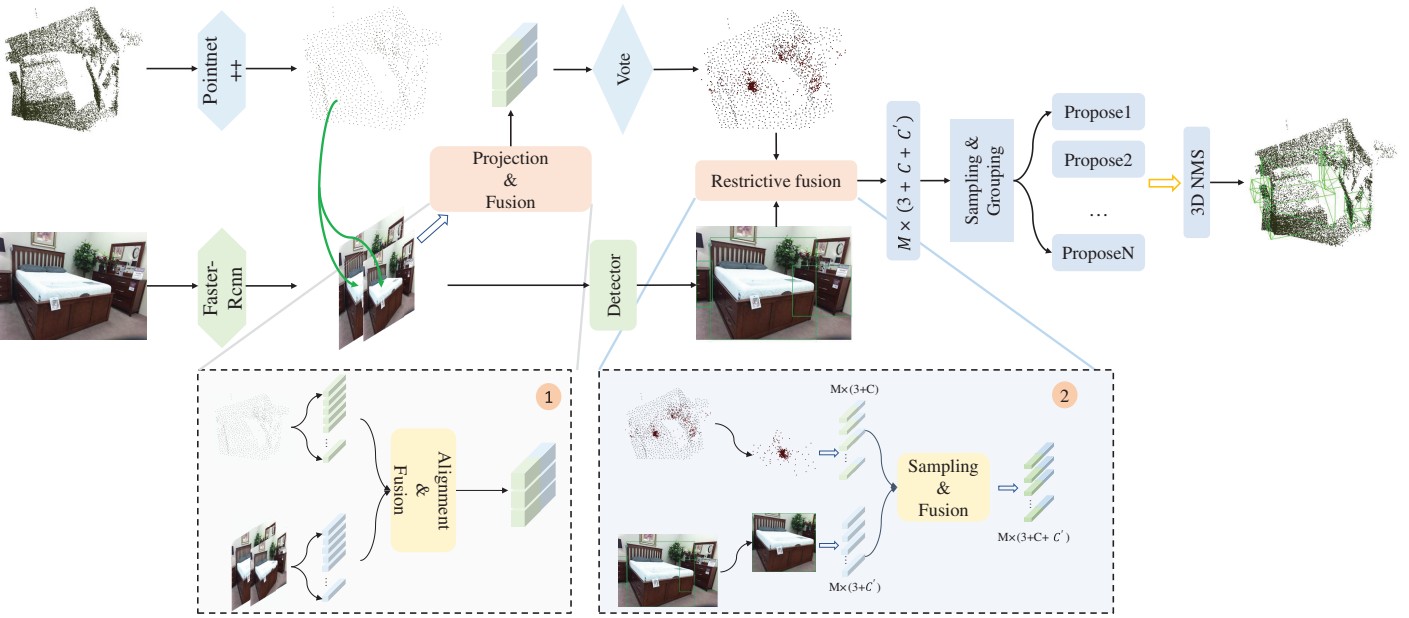

**Figure 2 3D object detection pipeline for TSFF.** The original point cloud and the RGB image are used as inputs for two branches: seed point feature extraction of the original point cloud with PointNet++ (*Qi et al., 2017*) as the backbone network, optimization processing of the voting points using the projection fusion module, selective sampling and feature fusion of reliable voting points with the help of the constraints fusion module, enhancement of the expression ability of the point features, and finally the final 3D bounding box using the region aggregation and the 3D NMS to predict the final 3D bounding box. Figure source credit: SUN RGB-D dataset.

## Vote mapping module

Point clouds and RGB images represent two distinct data modalities that differ significantly at the data representation level. To effectively integrate data from these two modalities, a matrix transformation must be applied to one of the data types to achieve alignment between them. When capturing 3D scene data using a sensor, we can simultaneously obtain RGB images from the same viewpoint along with the corresponding camera parameters. By utilizing the camera coordinate system as an intermediary and employing the camera's intrinsic matrix $K$ and rotation matrix $R_t$, we can achieve the transformation between 3D and 2D coordinates, as illustrated in Eq. (1):

$$X_I = M \times x_p;$$

(1)

M is the mapping matrix which can be specifically denoted as $K \times R_t$, where the projected 2D coordinates are denoted as $X_I = (u, v)$, the raw point cloud coordinates are denoted as $x_p = (x, y, z)$.

As depicted in Fig. 1C, the final scanned point cloud of an object's surface is often sparse and incomplete, a result of object occlusion and device defects. To estimate the approximate centroid of the target object from this sparse point cloud, we utilize the voting mechanism of VoteNet. Figure 3 illustrates the voting mapping module we have developed. Unlike the original voting mechanism, which only biases the point cloud features, our approach aims to provide the seed points with additional reference factors

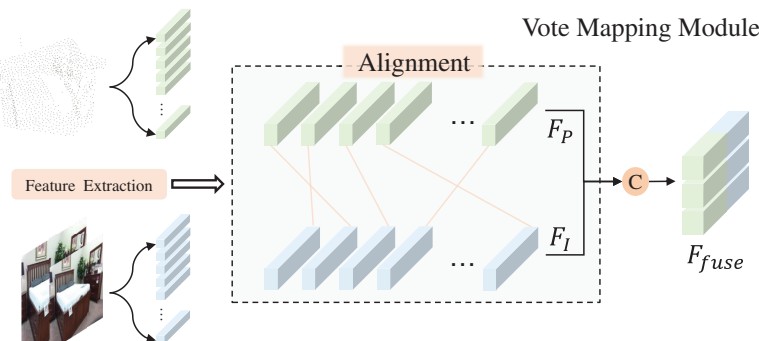

**Figure 3** **Vote mapping module.** Figure source credit: SUN RGB-D dataset.

during the bias stage. This is achieved by fusing the previously extracted image features $F_I$ with the point features $F_P$, as demonstrated in Eq. (2).

$$F_{fuse} = Fusion(F_P + F_I); \tag{2}$$

By utilizing seed points that are enhanced with image semantic features as inputs to the voting module, we can secure a higher number of object-related points during the voting bias phase. Consequently, the central voting point obtained is more representative of the actual object. The conclusion will be confirmed by the ablation experiments in the following content.

## Constraint fusion module

Based on the representational effects of point clouds, they can be divided into foreground point clouds and background point clouds. However, in the 3D scene represented by point cloud, the background points occupy a large portion of the data volume. The voting mechanism to add bias prompts the seed points to be offset to the object center, however, there still exists the problem of the interference of the background points. In order to deal with the interference of background points, the constrained fusion module (CFM) was developed.

As shown in Fig. 4, under the action of image branching, objects in RGB images can be represented by 2D bounding boxes. We project the obtained voting points onto the image again through the mapping matrix (Eq. (1)), then we sample the voting points using the 2D Bounding box as a constraint. When the coordinates of the mapped pixel fall within the object box, the corresponding polling point is recorded and retained, and the polling point will be ignored, referring to Eq. (3).

$$P^* = M \times Constraint(x^*, \gamma); \tag{3}$$

$P^*$ stands for vote points after sampling, and $Constraint(x^*, \gamma)$ represent limitations of the 2D bounding box to the projected coordinates. With $x^*$ indicating the coordinates of the projected point and $\gamma \in (0, 1)$ indicating whether or not it falls inside the box.
Constraint Fusion Module

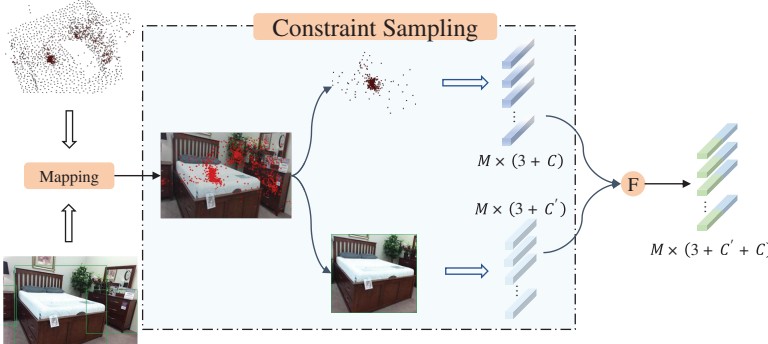

**Figure 4 Constraint fusion module.** Figure source credit: SUN RGB-D dataset.

Considering the two-dimensional characteristics of RGB, the occlusion between objects is unavoidable. Cross overlapping may occur between 2D bounding boxes, when the pixel coordinates converted from 3D coordinates happen to fall in the overlapping region, their attribution become problematic. In the CFM, we assign attribution labels to each mapped coordinate. When it falls in the overlapping region, the correlation of the coordinate is copied and backed up, with different labels to facilitate the subsequent differentiation. Meanwhile, since the biases added in the voting stage can cause part of the point cloud to fall outside the image boundaries during projection, we have also added an additional step of normalizing the coordinates of the projected points to constrain them to the image region.

## Loss function

Our loss design references VoteNet (*Qi et al., 2019*), which includes voting loss, object loss, 3D bounding box estimation loss, and semantic categorization loss. Specifically, it can be expressed as Eq. (4)

$$L = L_{vote} + w_1 \cdot L_{obj} + w_2 \cdot L_{sem} + L_{box};$$ (4)

## EXPERIMENTS

In this section, we first provide a brief overview of the characteristics of the SUNRGB-D dataset (*Song, Lichtenberg & Xiao, 2015*). We then present a visualization comparing the detection results of our method with those of the baseline within the same scene. Finally, we present the results of ablation studies to illustrate the contribution of different modules to the overall architecture and to demonstrate the robustness of our design modules.

## Datasets and comparing

**Dataset:** The SUNRGB-D dataset is an RGB-D image dataset specifically designed for 3D scene understanding. It is curated and expanded from SUN3D (*Xiao, Owens & Torralba, 2013*), NYU Depth v2 (*Silberman et al., 2012*), and Berkeley B3DO (*Janoch et al., 2013*), culminating in a collection of 10,335 indoor images depicting various scenes. Each image is

**Table 1 Briefly 3D object detection results on SUN RGB-D.**

| SUNRGB-D | Input | mAP@0.25 |
|---|---|---|
| DSS (*Song & Xiao, 2016*) | Geo+RGB | 42.1 |
| COG (*Ren & Sudderth, 2016*) | Geo+RGB | 47.6 |
| 2D-Driven (*Lahoud & Ghanem, 2017*) | Geo+RGB | 45.1 |
| PointFusion (*Xu, Anguelov & Jain, 2018*) | Geo+RGB | 45.4 |
| F-PointNet (*Qi et al., 2018*) | Geo+RGB | 54.0 |
| EpNet (*Huang et al., 2020*) | Geo+RGB | 59.8 |
| PiMAE † (*Chen et al., 2023*) | Geo+RGB | 59.4 |
| votenet (*Qi et al., 2019*) | Geo | 57.7 |
| MLCVNet (*Xie et al., 2020*) | Geo | 59.8 |
| H3DNet (*Zhang et al., 2020*) | Geo | 60.1 |
| BRNet (*Cheng et al., 2021*) | Geo | 61.1 |
| DAVNet (*Liang, An & Ma, 2022*) | Geo | 60.3 |
| SCNet (*Wei et al., 2023*) | Geo | 60.8 |
| Ours | Geo+RGB | 61.3 |

Note:
Geo, Geometric Features for Point Clouds; RGB, RGB images. PiMAE † is a pre-training framework and 3DETR is used a downstream task for 3D object detection.

accompanied by corresponding depth information, camera parameters, and object labeling information. Of these, 5,285 images constitute the training set, with the remainder serving as the test set. The dataset encompasses annotations for a total of 37 object classes. Utilizing depth maps, the 3D point cloud information of the scene can be generated, with each point featuring a semantic label and object bounding box information. The alignment between the RGB image and the depth channel is achieved using the camera parameters. VoteNet was used as the baseline, with which to train our model and report prediction results for the ten most prevalent categories in indoor scenes. Given that the ScanNet dataset includes instances where a single point cloud scene encompasses multiple image views and considering the complexity of multi-view processing, our method did not undergo evaluation on this aspect for validation.

**Comparison:** As shown in Table 1, we evaluated TSFF using the SUNRGB-D dataset and compared it with previous indoor 3D object detection methods. We categorized the experimental results based on whether RGB image features were utilized after considering the differences in input data used by different indoor 3D object detection methods. Earlier experiments (*Ren & Sudderth, 2016*; *Song & Xiao, 2016*; *Lahoud & Ghanem, 2017*; *Xu, Anguelov & Jain, 2018*; *Qi et al., 2018*) often fused the two modalities during the proposal stage. EpNet (*Huang et al., 2020*) attempted to integrate image features at the early stage of feature learning through point guidance. PiMAE (*Chen et al., 2023*) utilized a multimodal pre-training framework for fine-tuning downstream tasks to achieve better detection results.

Additionally, we compared TSFF with recent outstanding single-modal 3D object detection results and from the final detection results, it is evident that leveraging image features greatly benefits 3D object detection. MLCNet (*Xie et al., 2020*) takes point clouds

**Table 2  3D object detection results from the SUN RGB-D v1 val set.** Values in bold indicate the highest precision within the respective category. PiMAE † is a pre-training framework, and 3DETR is used as a downstream task for 3D object detection.

| Methods | Bathtub | Bed | Bookshelf | Chair | Desk | Dresser | Nightstand | Sofa | Table | Toliet | mAP@0.25 |
|---|---|---|---|---|---|---|---|---|---|---|---|
| DSS | 44.2 | 78.8 | 11.9 | 61.2 | 20.5 | 6.4 | 15.4 | 53.5 | 50.3 | 78.9 | 42.1 |
| COG | 58.3 | 63.7 | 31.8 | 62.2 | **45.2** | 15.5 | 27.4 | 51.0 | 51.3 | 70.1 | 47.6 |
| 2D-Driven | 43.5 | 64.5 | 31.4 | 48.3 | 27.9 | 25.9 | 41.9 | 50.4 | 37.0 | 80.4 | 45.1 |
| PointFusion | 37.3 | 68.6 | 37.7 | 55.1 | 17.2 | 23.9 | 32.3 | 53.8 | 31.0 | 83.8 | 45.4 |
| F-PointNet | 43.3 | 81.1 | 33.3 | 64.2 | 24.7 | 32.0 | 58.1 | 61.1 | 51.1 | 90.9 | 54.0 |
| EpNet | 75.4 | 85.2 | 35.4 | 75.0 | 26.1 | 31.3 | 62.0 | 67.2 | 52.1 | 88.2 | 59.8 |
| PiMAE † | **80.3** | 85.4 | 30.4 | 69.0 | 28.2 | 33.0 | 62.8 | 62.5 | 48.9 | **93.8** | 59.4 |
| VoteNet | 74.4 | 83.0 | 28.8 | 75.3 | 22.0 | 29.8 | 62.2 | 64.0 | 47.3 | 90.1 | 57.7 |
| MLCVNet | 79.2 | 85.8 | 31.9 | 75.8 | 26.5 | 31.3 | 61.5 | 66.3 | 50.4 | 89.1 | 59.8 |
| H3DNet | 73.8 | 85.6 | 31.0 | 76.7 | 29.6 | 33.4 | 65.5 | 66.3 | 50.8 | 88.2 | 60.1 |
| BRNet | 76.2 | **86.9** | 29.7 | **77.4** | 29.6 | 35.9 | 65.9 | 66.4 | 51.8 | 91.3 | 61.1 |
| DAVNet | 78.9 | 84.6 | 29.4 | 77.1 | 27.5 | 32.2 | 65.0 | 66.4 | 52.1 | 90.0 | 60.3 |
| SCNet | 74.5 | 85.9 | 31.7 | 76.9 | 30.3 | 34.2 | 67.1 | 66.9 | **52.3** | 88.6 | 60.8 |
| Ours | 73.4 | 86.6 | **38.7** | 75.3 | 25.7 | **39.5** | **66.8** | **70.5** | 47.3 | 89.2 | **61.3** |

Note:
  The evaluation metric is the average precision with 3D IOU threshold as 0.25.

as input and introduces different levels of contextual information in the voting and classification stages, along with a global scene context module to learn global scene context. DAVNet (*Liang, An & Ma, 2022*) emphasizes object refinement and localization quality estimation, refining discriminative features through adaptive perception fields to provide reliable localization confidence. SCNet (*Wei et al., 2023*) focuses on the direct semantic properties of point clouds and the consistency of geometric clues, achieving more robust detection results by analyzing the relationship between proposals and semantic segmentation points.

## Results and analytics

**Result:** Table 2 presents the 3D object detection results for ten common categories on the SUNRGB-D dataset. We employ VoteNet as the baseline for experimental comparison. Our method demonstrates an improvement of 3.6 mAP over the baseline, utilizing mAP@0.25 as the evaluation criterion, and achieves notable enhancements in detection accuracy for several categories (bookshelf: +9.9% AP, dresser: +9.7% AP, sofa: +6.5% AP). In comparison with other state-of-the-art methods, TSFF exhibits superior performance in detecting objects within categories such as bookshelf, dresser, nightstand, and sofa. Given that these object categories frequently co-occur with other categories in realistic scenarios, there is a propensity for misdirection during the clustering operation. The benefit of employing 2D bounding boxes to sample voting points in the constraint fusion module is effectively validated here.

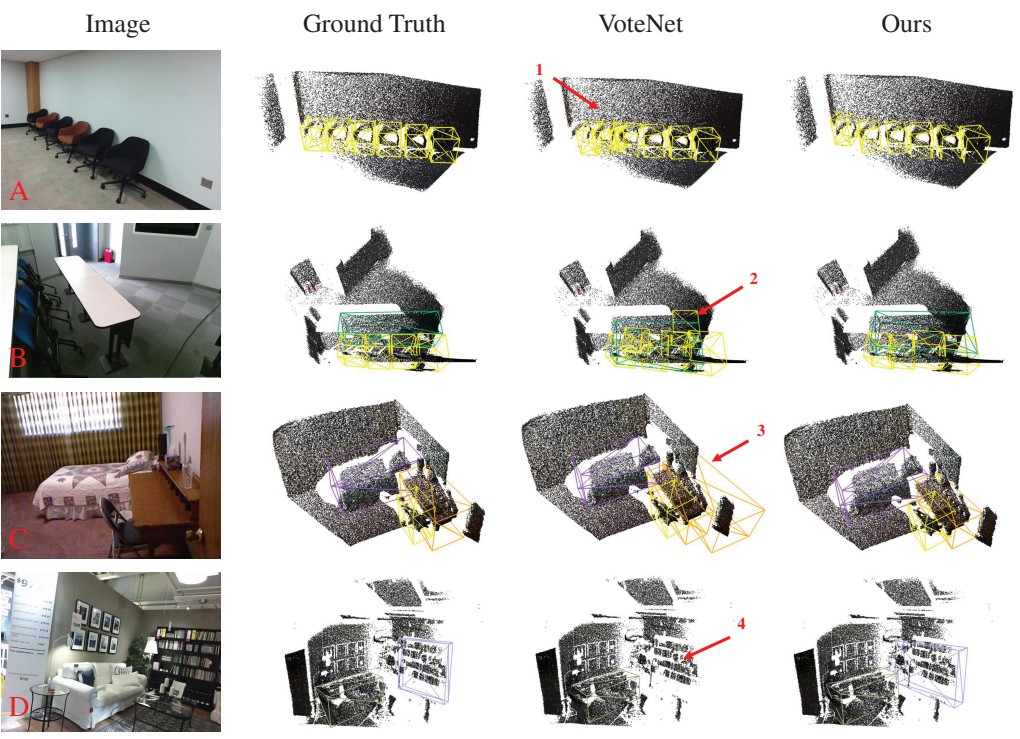

**Figure 5** **Visualization of the 3D object detection results from different methods on the SUNRGB-D dataset.** Figure source credit: SUN RGB-D dataset.     

**Visual analytics:** To visually demonstrate the improvement in detection accuracy of our method compared to the baseline, we visualize the detection results of Ground Truth (GT), VoteNet, and our TSFF in Fig. 5, accompanied by indications of detection errors. In the four scenarios shown in Fig. 5, scenarios A and B exhibit small and densely packed chair objects. During the process of clustering and grouping the voting points, the clustering centers are susceptible to interference from surrounding objects, leading to errors in center point predictions. As observed in the first and second instances of errors, instances of false positives are evident in both scenarios. Scenarios C and D are relatively more complex than the previous, with more background noise around the detected objects. By observing the third error instance, it can be seen that the presence of background noise points causes the VoteNet model to incorrectly identify them as desks. In the fourth error instance, the bookshelf is influenced by wall noise, and the object surface point cloud is mistakenly identified as background noise, resulting in a missed detection. In contrast, our method fully exploits the excellent semantic representation of image features, resulting in consistent detection results with the GT.

Figure 6 illustrates the distribution of voting points projected onto the 2D image after being processed by the CFM, compared with the distribution of voting points without using this module. From the figure, it can be seen that, with the same number of projection points, the voting points processed by the CFM are more concentrated around the center of the detected objects compared to those without the CFM. This suggests that, with the

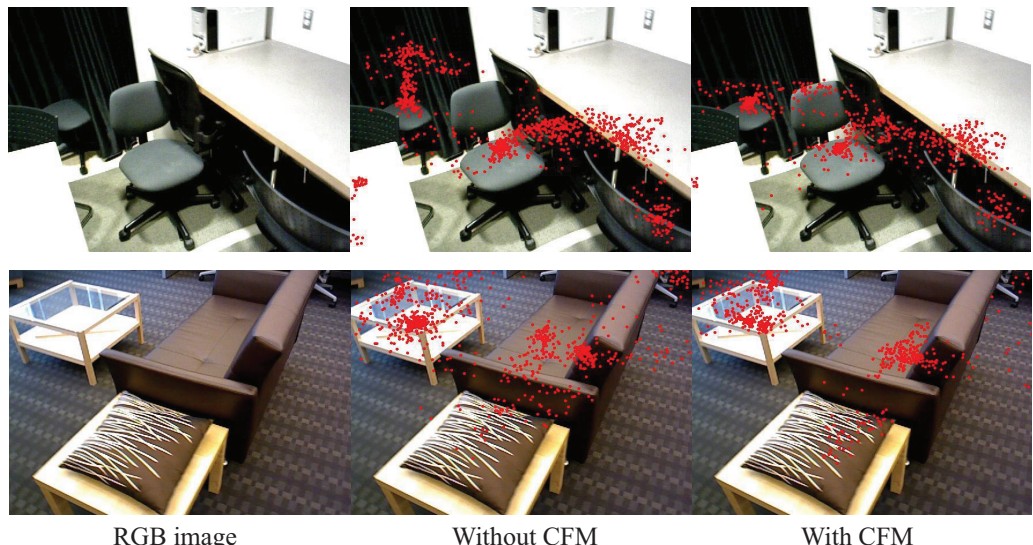

|                | RGB image | Without CFM | With CFM |

**Figure 6 Visualization of projected points after constraint projection module processing.** Figure source credit: SUN RGB-D dataset.

|                | Image | Ground Truth | VoteNet | Ours |

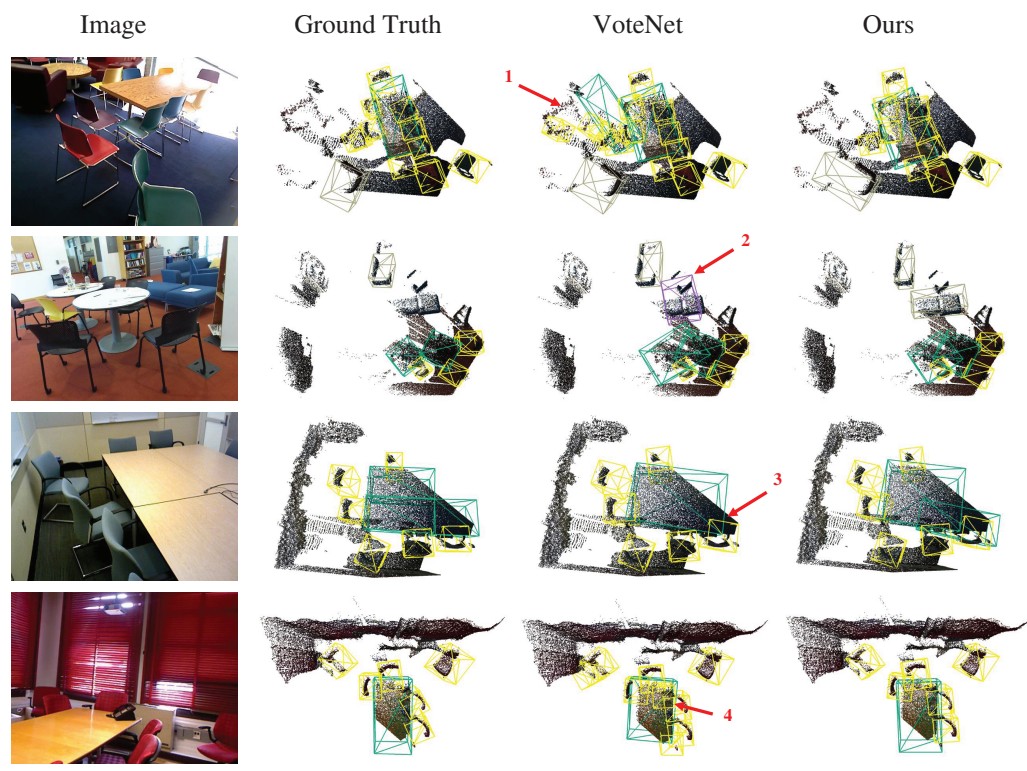

**Figure 7 Visualization of 3D object detection results in complex scenes.** Figure source credit: SUN RGB-D dataset.

**Table 3 Alabtion study for our TSFF.**

| Baseline | VMM | CFM | mAP@0.25 |
|---|---|---|---|
| ✓ | ✗ | ✗ | 57.7 |
| ✓ | ✓ | ✗ | 58.3 |
| ✓ | ✗ | ✓ | 56.6 |
| ✓ | ✓ | ✓ | 61.3 |

**Note:**
Baseline stands for VoteNet.
✓ The module was selected.
✗ The module was not selected.

assistance of 2D bounding boxes in the CFM, we can effectively filter out voting points outside the target region, thereby suppressing background noise interference.

We also demonstrate the effectiveness of our method in complex scenarios (see Fig. 7). The detection results indicate that our method surpasses VoteNet in terms of accuracy. Additionally, while maintaining consistency with the ground truth (GT), our method accurately detects objects that were missed in the dataset annotations. In contrast, although VoteNet successfully detected these objects, it incorrectly classified them as 'bed', highlighting the robustness of our method in complex scenes.

## Alabtion study

In this subsection, to further analyze the detection performance of our designed TSFF model, we conduct experimental analyses on the VMM and CFM separately, discussing the contributions of different modules to the model's performance. Additionally, we visualize the voting point clouds processed by CFM to demonstrate the module's handling of background points. All of our experiments are conducted using the SUNRGB-D dataset and evaluated using mAP@0.25.

As presented in Table 3, we conducted experiments to assess the impact of adding the VMM and the CFM individually to the baseline, with VoteNet serving as the baseline, and compared these results with those of the complete module. The addition of a single module, while not yielding a significant improvement, demonstrates that the vote mapping module aids in acquiring better voting points. Conversely, the inclusion of only the constraint fusion module did not produce satisfactory results. This indicates a synergistic relationship between the two modules, where both the ability to secure high-quality voting points and the utilization of superior image features significantly contribute to the final detection outcomes.

## CONCLUSIONS

In this research, we focus on the instability problems of indoor object detection in the presence of sparse point clouds on the object surface and severe interference from background noise points. Our TSFF model enhances the point cloud features with image features containing rich semantic information and aids 3D object recognition with the help of a 2D image detection frame. Our results show that the use of a 2D bounding box as a

constraint limiting the sampling of voting points can effectively mitigate the interference of background points, thus obtaining more robust detection results.

## ACKNOWLEDGEMENTS

We sincerely thank all the faculty and students who contributed to this study. Our experiments are based on the SUNRGB-D dataset, and we greatly appreciate their outstanding contributions to the field.

### Funding

This work was supported by the Natural Science Foundation of Xiamen, China (No. 3502Z202373036), the National Natural Science Foundation of China (No. 42371457), the Open Competition for Innovative Projects of Xiamen, China (No. 3502Z20231038) and the Key Project of Natural Science Foundation of Fujian Province, China (No. 2022J02045). There was no additional external funding received for this study. The funders had no role in study design, data collection and analysis, decision to publish, or preparation of the manuscript.

### Grant Disclosures

The following grant information was disclosed by the authors:
Natural Science Foundation of Xiamen, China: 3502Z202373036 and 42371457.
Open Competition for Innovative Projects of Xiamen, China: 3502Z20231038.
Natural Science Foundation of Fujian Province, China: 2022J02045.

### Competing Interests

The authors declare that they have no competing interests.

### Author Contributions

- Guoqing Jiang conceived and designed the experiments, performed the experiments, analyzed the data, performed the computation work, prepared figures and/or tables, authored or reviewed drafts of the article, and approved the final draft.
- Saiya Li performed the experiments, performed the computation work, authored or reviewed drafts of the article, and approved the final draft.
- Ziyu Huang performed the experiments, performed the computation work, authored or reviewed drafts of the article, and approved the final draft.
- Guorong Cai conceived and designed the experiments, analyzed the data, prepared figures and/or tables, and approved the final draft.
- Jinhe Su conceived and designed the experiments, analyzed the data, prepared figures and/or tables, and approved the final draft.

### Data Availability

The data, source code, pre-trained model and experimental results are available at Zenodo: Jiang, G. (2024). code files. Zenodo. https://doi.org/10.5281/zenodo.12205533. The SUN RGB-D dataset is available at Princeton: https://rgbd.cs.princeton.edu.

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
