# Peer review of "TSFF: a two-stage fusion framework for 3D object detection"

_PeerJ Computer Science, doi:10.7717/peerj-cs.2260_

## Round 0.1 · original submission · Major Revisions

The reviewers agree on the validity of the work and the importance of the research topic. However, improvements are still required before publication. In particular, the reviewers suggest a comparison with more recent approaches. Please consult the reviews for detailed comments.

Reviewer 1 ·

Basic reporting

The article is often not clear, with poor use of the English language.
Acronyms are used without an explanation, e.g. line 148, line 154.
At times some sentences do not seem to make sense, e.g. line 159 to 161: why are the authors talking about LIDARs if tthis approach is based on RGBD data?

The introduction and related works only cite very recent works, while in the experiments section authors cite and compare their approach to older works, which are almost completely disjointed from the works cited in the previous sections.

The structure of the article conforms to the `standard sections’ suggested by PeerJ.
The authors shared the raw data and code, however no instructions for replicating the experiments are given.

The article could represent appropriate unit of publication.

Experimental design

There seems to be a research question, e.g. how to integrate 3D and 2D data together for the task of 3D object detection, however it is not clear what is the novelty compared to the state of the art methods.

Some improvement w.r.t. the SOTA methods is shown, however the authors do not compare themselves with more recent work.

Validity of the findings

no comment

Cite this review as

Reviewer 2 ·

Basic reporting

This paper proposes a two-stage fusion 3D object detection framework combined with point clouds and RGB images. They put forward Vote Mapping Module to align the fuse the point features and image features. And Constraint Fusion Module is utilized for the second fusion of features. The whole model is trained with the loss function which consists of voting loss, object loss, 3D bounding box estimation loss and semantic categorization loss. They design experiments to verify(1) improved accuracy of object detection evaluated on mAP.25 (2) effectiveness of VMM and CFM.

Strengths of the paper:

1. The paper is organized clearly for readers to understand the paper's motivation and conclusion.

2. The method is simple but effective to improve the performance.

3. Visualization of the detection result validates the performance.

4. Ablation study shows how exactly different components contribute to the final improvement.

Experimental design

no comment

Validity of the findings

Suggestions to improve:

1. The method must be evaluated more thoroughly with more datasets. Currently only one indoor dataset is utilized to exhibit the improvement.

2. Faster RCNN has been put forward in 2015, so it is worthwhile to try new models for the feature extraction of image branch such as SAM, which may bring potential improvement. As well as the point branch, other architectures like DGCNN, PointTransformer could be explored for more comprehensive experiments.

3. The bounding box in Figure 5 can be represented in bold to make it more clear for readers.

4. In my understanding, the fusion of features is realized by concatenation. Other novel manners like attention-based fusion could be experimented for further improvement.

5. The authors can provide more analysis of how CFM refines the selection of voting points and reduces the interference from background points in experiments.

Cite this review as

---

## Round 0.2 · accepted · Accept

The authors have addressed the reviewers' comments, including those that required additional experiments. Concerning the English writing and overall presentation, which was a significant concern for the reviewers, the paper has undergone an extensive review that has greatly improved its readability, thanks in part to the PeerJ editorial service as acknowledged by the authors.

Another major concern was the experimental section. To address this, the authors added a comparison with a recent approach, PiMAE, as well as other older methods. After this addition I find this comparison to be sufficiently comprehensive, especially considering the presence also of an ablation study.

As editor of PeerJ CS, I have personally assessed the revised paper since the two previous reviewers declined to review it again. I consider the paper ready for publication.